# On the Generalization Capacities of MLLMs for Spatial Intelligence

**Gongjie Zhang**[1,2*] **Wenhao Li**[3*] **Quanhao Qian**[1,2] **Jiuniu Wang**[1,2] **Deli Zhao**[1,2] **Shijian Lu**[3] **Ran Xu**[1,2✉]

[1]DAMO Academy, Alibaba Group     [2]HuPan Lab     [3]Nanyang Technological University

Project Page: https://github.com/Vegetebird/CA-MLLM

## Abstract

Multimodal Large Language Models (MLLMs) that directly process RGB inputs for tasks like 3D localization and navigation have shown remarkable potential. However, we argue that these "RGB-only" approaches are fundamentally flawed in their ability to generalize across cameras. By ignoring camera parameters, they entangle an object's physical properties with the camera's perspective, creating an irresolvable ambiguity. We show this leads MLLMs to overfit to the training camera distribution, rather than learning true and generalizable 3D geometric principles. To address this, we propose *Camera-Aware MLLM framework* for spatial MLLMs. It learns generalizable spatial reasoning by: *(i)* injecting camera intrinsics via a dense embedding that conditions each visual token; *(ii)* introducing a camera-aware data augmentation strategy that synthetically varies camera parameters, forcing the model to disentangle camera properties from scene content; and *(iii)* distilling geometric priors from a 3D vision foundation model. Extensive experiments demonstrate that camera-aware MLLMs substantially outperform their naive counterparts, particularly in cross-camera generalization tests on spatially-grounded tasks, indicating that camera-awareness is not only beneficial but also a prerequisite for robust and generalizable spatial intelligence in MLLMs.

## 1 Introduction

Multimodal Large Language Models (MLLMs) are rapidly advancing the frontier of spatial intelligence, enabling AI that can perceive and reason about 3D environments through natural language. While early approaches often relied on explicit 3D representations such as point clouds (Xu et al., 2024; Hong et al., 2023; Wang et al., 2024), a prominent paradigm has emerged: feeding RGB images or videos directly into MLLMs for end-to-end training on spatial tasks like 3D localization, depth estimation, relational understanding, and navigation (Zheng et al., 2025; Zhang et al., 2025; Chen et al., 2024a). This RGB-centric methodology, not relying on any 3D data, has shown impressive results, suggesting that MLLMs can implicitly learn spatial principles from 2D data.

However, we identify a fundamental flaw in this paradigm: the omission of camera intrinsic parameters. This oversight creates an irresolvable geometric ambiguity that undermines cross-camera generalization. As illustrated in Fig. 1, in the pinhole camera model, a fronto-parallel object of physical height $H$ at depth $Z$ projects to an image height

$$h_{\mathrm{proj}} = \frac{f\,H}{Z}.$$ (1)

Equation 1 induces an equivalence class of image observations: for any $\lambda > 0$,

$$(f, H, Z) \sim (\lambda f, H, \lambda Z) \sim (f, \lambda H, \lambda Z),$$ (2)

all yield the same $h_{\mathrm{proj}}$. Thus, without camera intrinsics, an RGB-only MLLM cannot disambiguate a nearby small object from a distant large one, nor can it separate depth changes from focal-length (zoom) changes. The ambiguity is exacerbated by principal-point shifts and pixel aspect ratio: even when two images "look" similar, their per-pixel rays differ if camera intrinsics differ, leading models to learn camera-specific shortcuts instead of generalizable 3D principles. Prior work in monocular metric depth estimation (Yin et al., 2023; Piccinelli et al., 2024) has shown that canonicalizing intrinsics or injecting them explicitly is crucial for cross-camera generalization; we believe that the same lesson must carry over to MLLM-based spatial reasoning.

---

*Equal Contribution.    ✉Corresponding Author.

To bridge this gap, we introduce the *Camera-Aware MLLM framework*, designed to make spatial reasoning explicitly camera-aware via three core technical innovations. *First*, we develop a dense camera embedding mechanism that conditions every visual token on the corresponding camera's ray direction derived from intrinsic parameters, enabling the model to reason about the geometric relationship between pixels and 3D

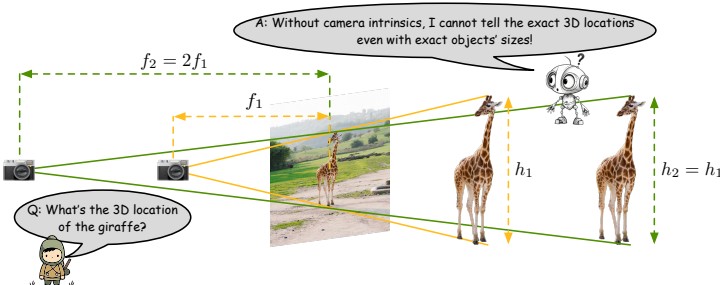

Figure 1: Illustration of the inherent geometric ambiguity in RGB-only spatial reasoning. The same 2D image can result from different 3D scenes: a nearby object with a wide-angle lens can appear identical to a distant object with a telephoto lens. This ambiguity makes generalizable 3D localization from a single RGB image an ill-posed problem, especially when camera intrinsics are unknown.

space. *Second*, recognizing that existing 3D datasets often lack sufficient camera diversity as compared with 2D datasets, we propose a camera-aware data augmentation strategy. By synthetically varying camera intrinsics and applying the corresponding geometric transformations to the visual input, we compel the model to disentangle camera properties from scene content. *Third*, to further ground our model in robust geometric principles, we leverage a pre-trained 3D vision foundation model, trained on millions of RGB-depth pairs across diverse cameras, to distill geometric priors, enriching the MLLM's understanding while maintaining an efficient, RGB-only inference pipeline.

To validate our framework, we conduct extensive experiments on the cross-camera generalization capability of spatial MLLMs. Our experiments reveal a stark performance gap: camera-agnostic baselines fail catastrophically on out-of-distribution cameras, whereas our camera-aware models maintain robust performance. These findings demonstrate that explicit camera awareness is crucial for attaining reliable spatial intelligence in MLLMs.

In summary, our contributions are threefold. *First*, we provide an in-depth analysis revealing the inherent geometric ambiguity in RGB-only spatial reasoning, both theoretically and empirically, and demonstrate that without camera intrinsics, MLLMs cannot learn true and generalizable 3D geometric principles. *Second*, we propose the Camera-Aware MLLM framework, the first architecture to explicitly address geometric ambiguity in spatial reasoning through dense camera embeddings, geometric prior distillation, and camera-aware augmentation. *Third*, extensive experiments verify the effectiveness of our method, establishing camera-awareness as a prerequisite for generalization and offering a clear blueprint for future research. Our work argues for a shift from merely processing pixels to understanding the geometric principles governing their formation, steering the field toward truly generalizable spatial AI.

## 2 RELATED WORK

**Multimodal Large Language Models (MLLMs).** MLLMs extend the powerful reasoning and language capabilities of LLMs to visual modalities like images (Li et al., 2022; 2023; Alayrac et al., 2022; Liu et al., 2023; 2024) and videos (Zhang et al., 2023; Lin et al., 2023). By aligning visual encoders with large language models, they excel at a diverse range of vision-language tasks, including object grounding (Peng et al., 2023; Zhang et al., 2024a), image captioning (Lee et al., 2024; Hua et al., 2025), visual question answering (Jiang et al., 2025b; Kuang et al., 2025), and complex reasoning (Chen et al., 2024b; Huang et al., 2025). Current research continues to scale models, expand their capabilities to video and fine-grained understanding, and develop more sophisticated architectures (Hurst et al., 2024; Comanici et al., 2025; Bai et al., 2025), establishing MLLMs as a foundational technology in AI.

**MLLMs for Spatial Intelligence.** The impressive capabilities of MLLMs have been naturally extended to spatial intelligence for understanding 3D environments. One line of work directly processes 3D representations, such as point clouds, for tasks like 3D question answering and localization (Hong et al., 2023; Xu et al., 2024; Wang et al., 2024; Chen et al., 2024c; Guo et al., 2025; Miao et al., 2025). However, the relative scarcity of large-scale 3D data has motivated an alternative, RGB-only paradigm. In this approach, standard 2D MLLMs are trained to implicitly grasp

Table 1: Generalization failure of camera-agnostic MLLMs. 3D object detection performance drops when trained on mixed data sources or evaluated on resized images, exposing a fundamental lack of robustness and generalization.

| Model | Train Data | Evaluate Data | 31 common classes | | |
|---|---|---|---|---|---|
| | | | $P_{0.25}$ | $R_{0.25}$ | $F1_{0.25}$ |
| Qwen2.5-VL 3B | ScanNet | ScanNet-val | 47.5 | 44.2 | 45.7 |
| | ScanNet ARKitScenes 3RScan Matterport3D SUN-RGBD Objectron | ScanNet-val | 38.4 | 33.3 | 35.4 |
| | ScanNet | ScanNet-val x0.8 | 26.4 | 22.9 | 24.3 |
| | ScanNet | ScanNet-val x1.2 | 33.0 | 30.5 | 31.6 |
| VG-LLM 4B | ScanNet | ScanNet-val | 48.3 | 45.0 | 46.5 |
| | ScanNet ARKitScenes 3RScan Matterport3D SUN-RGBD Objectron | ScanNet-val | 49.5 | 43.6 | 46.0 |
| | ScanNet | ScanNet-val x0.8 | 26.7 | 25.0 | 25.8 |
| | ScanNet | ScanNet-val x1.2 | 34.7 | 32.1 | 33.2 |

• 'ScanNet-val x0.8' refers to ScanNet validation set images being resized by a factor of 0.8.

spatial concepts directly from images or videos, offering greater data scalability and showing significant promise (Chen et al., 2024a; Zhang et al., 2025; Zheng et al., 2025). Vision-Language-Action (VLA) models (Zitkovich et al., 2023; Kim et al., 2024; Jiang et al., 2025a; Qian et al., 2025) for robotics and autonomous driving further highlight the potential of MLLMs to generate real-world actions from visual input. Our work operates within this RGB-only paradigm but identifies and resolves a critical oversight: the systemic neglect of camera intrinsic parameters, which fundamentally limits cross-camera generalization capabilities.

**Monocular Metric Depth Estimation (MMDE).** MMDE, which predicts per-pixel metric depth from a single RGB image, is a classic ill-posed problem due to the inherent scale ambiguity of 2D projection. While early methods (Eigen et al., 2014; Patil et al., 2022; Bhat et al., 2021; Piccinelli et al., 2023) failed to generalize across cameras, breakthrough works like Metric3D (Yin et al., 2023; Hu et al., 2024) and UniDepth (Piccinelli et al., 2024; 2025) achieved cross-camera generalization by explicitly considering camera intrinsics. Their key strategies involve either canonicalizing inputs with known intrinsics or conditioning the network on intrinsics, thereby disentangling camera geometry from scene content. These works have established that camera-awareness is a prerequisite for robust generalization. Our work draws inspiration from this insight, extending the principle from the specialized task of depth estimation to the broader domain of spatial reasoning in MLLMs and leveraging a universal MMDE model (Piccinelli et al., 2025) to distill geometric priors.

## 3 BRITTLENESS OF CAMERA-AGNOSTIC MLLMS IN SPATIAL REASONING

### 3.1 THE CHALLENGE OF SPATIALLY-GROUNDED TASKS

We categorize spatial reasoning tasks into two types: *(i)* **Relational tasks** involve qualitative spatial relationships (*e.g.*, "Is the cup to the left of the monitor?") and do not require precise measurements; *(ii)* **Spatially-grounded tasks** demand quantitative 3D understanding with queries or answers anchored to a coordinate frame (*e.g.*, "Where is the 3D location of the red chair?" or "Describe the object at (x,y,z).").

Mastering spatially-grounded tasks is crucial for embodied AI in applications like robotics and autonomous driving. Yet, even advanced models like GPT-4o (Hurst et al., 2024) and Gemini-2.5 (Comanici et al., 2025) struggle with reliable 3D grounding. Our work aims to bridge this critical gap, moving MLLMs beyond qualitative recognition towards the quantitative precision required for true spatial intelligence.

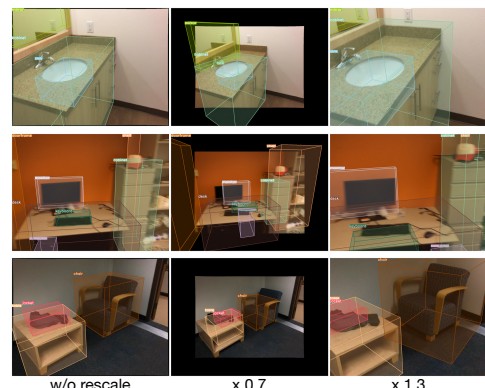

w/o rescale          x 0.7          x 1.3

Figure 2: Visualization of generalization failure in a camera-agnostic MLLM (finetuned Qwen2.5-VL). Simply resizing the input image during inference induces a systematic shift in 3D localization.

## 3.2 EMPIRICAL EVIDENCE OF GENERALIZATION FAILURE

While the challenge of spatially-grounded reasoning remains largely an open problem, pioneering works have shown promising results. Notably, VG-LLM (Zheng et al., 2025) has demonstrated that MLLMs can be trained/finetuned to reason about 3D space from video data, and generalist MLLMs like Qwen2.5-VL (Bai et al., 2025) can achieve decent results after targeted fine-tuning. In our initial experiments, we confirmed their effectiveness by training and evaluating VG-LLM (Zheng et al., 2025) and Qwen2.5-VL (Bai et al., 2025) for single-frame 3D object detection—the most basic spatially-grounded task—on the ScanNet dataset (Dai et al., 2017), establishing a strong single-dataset baseline as shown in Table 1. However, this success proves to be superficial when these models are pushed to generalize.

**Failure of Scaling-Up on Mixed-Source Datasets.** The core strength of MLLMs lies in their ability to scale up with diverse and large-scale data. However, our attempts to leverage this strength for spatial reasoning led to a counterintuitive outcome. As shown in Table 1, when we train baseline models on a large, aggregated datasets composed of multiple indoor scene collections, their performance on the ScanNet validation set drops. This suggests that the models are confused by the conflicting geometric signals from different camera sources, failing to generalize.

**Failure under Simple Geometric Transforms.** To isolate the cause of this fragility, we conducted a controlled experiment. We trained a model exclusively on ScanNet and tested it on the same dataset's images after applying a simple resize-and-pad (or resize-and-crop) transformation—a common preprocessing step. As shown in Table 1, performance collapsed under this simple transformation. This is not a minor degradation simply caused by resolution change; visualizations of the output (*e.g.*, Fig. 2) reveal that the predicted 3D locations become systematically and severely offset. This finding strongly suggests that the model has not learned generalizable geometric principles. Instead, it has severely overfit to the specific resolution of the training images, a property that proves to be brittle and non-generalizable. While Table 1 establishes the baseline failure on cross-camera generalization, in our experiments (Fig.6 and Table 5), we show that the proposed Camera-Aware MLLM framework greatly mitigates this failure in the more challenging cross-camera setting.

## 3.3 ANALYSIS: THE UNRESOLVED 3D GEOMETRIC AMBIGUITY

The empirical failures observed in Section 3.2 trace back to the geometric ambiguity inherent in any camera-agnostic approach. Without knowledge of camera intrinsics, a model cannot disentangle scene properties from camera properties, leading to overfitting on sensor geometry rather than learning true 3D principles.

**Theoretical Analysis: A Problem of Indistinguishability.** The relationship between a 3D world and its 2D projection is formally described by the pinhole camera model. As established in Eq. 1, the projected height of a fronto-parallel object, $h_{\mathrm{proj}}$, is a function of its physical height $H$, its depth $Z$, and the camera's vertical focal length $f_y$: $h_{\mathrm{proj}} = f_y H/Z$. This equation gives rise to an equivalence class of scenes that are indistinguishable from a single RGB image. We can formalize at least two critical types of ambiguity:

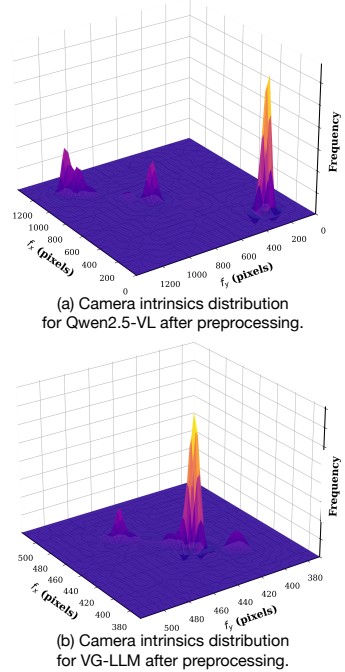

(a) Camera intrinsics distribution for Qwen2.5-VL after preprocessing.

(b) Camera intrinsics distribution for VG-LLM after preprocessing.

Figure 3: Multi-modal distribution of camera intrinsics in mixed datasets.

- **Focal-Depth Ambiguity:** A change in focal length is observationally equivalent to a change in depth. For any scaling factor $\lambda > 0$, the configuration $(\lambda f_y, H, \lambda Z)$ yields the same projected height as $(f_y, H, Z)$. A camera zooming in is indistinguishable from an object moving closer.
- **Size-Depth Ambiguity:** An object's physical size is confounded with its depth. The configuration $(f_y, \lambda H, \lambda Z)$ is indistinguishable from $(f_y, H, Z)$. A small, nearby object can project to the same size as a large, distant one.

While MLLMs can partially mitigate the size-depth ambiguity by leveraging strong priors about object sizes (*e.g.*, knowing a "chair" has a typical height $H$), this mechanism is critically de-

pendent on a stable, known focal length. The focal-depth ambiguity fundamentally undermines this prior-based reasoning. A camera-agnostic model is forced to assume a canonical focal length, $f_{y,\text{assumed}}$, implicitly learned from its training distribution. Any deviation in the test camera's $f_y$ from this assumption will systematically corrupt its depth and scale estimations. A formal derivation of these ambiguities is provided in Appendix A.

**Explaining the Failures.** The above theory provides an explanation for our empirical findings.

**1. Failure on Mixed-Source Datasets.** The performance degradation on aggregated datasets is a direct consequence of the model being exposed to a multi-modal distribution of camera intrinsics. As illustrated in Fig. 3, datasets like ScanNet (Dai et al., 2017), ARKitScenes (Baruch et al., 2021), 3RScan (Wald et al., 2019) and Matterport3D (Chang et al., 2017) each possess distinct and clustered focal length distributions. A camera-agnostic MLLM, attempting to learn a single $f_{y,\text{assumed}}$, is faced with conflicting geometric signals. It cannot converge to a coherent model, resulting in confusion and a net decrease in performance, as it averages over incompatible geometric worlds.

**2. Failure under Geometric Transforms (Image Resizing).** The resizing experiment provides the most direct proof of this flaw. Image resizing is not a mere change in resolution; it is an explicit transformation of the camera's intrinsic parameters. Resizing an image by a factor $s$ is mathematically equivalent to scaling the focal lengths and the principal point coordinates: $(f_x, f_y, c_x, c_y) \rightarrow (s \cdot f_x, s \cdot f_y, s \cdot c_x, s \cdot c_y)$. A model trained on original-resolution images has learned to operate with $f_{y,\text{train}}$. When presented with a resized image, its internal reasoning becomes:

$$Z_{\text{pred}} \approx \frac{f_{y,\text{assumed}} \cdot H_{\text{prior}}}{h_{\text{proj, resized}}} = \frac{f_{y,\text{train}} \cdot H_{\text{prior}}}{(s \cdot f_{y,\text{train}} \cdot H_{\text{physical}})/Z_{\text{physical}}} \approx \frac{Z_{\text{physical}}}{s}.$$

This predicts a systematic error: the model's depth estimate will be inversely proportional to the resize factor $s$. This explains the huge performance drop in Table 1 and the systematic localization offset as observed in Fig. 2.

In summary, **the failure of camera-agnostic MLLMs is not a limitation of model scale or architecture, but stems from a fundamental information deficit: the absence of camera intrinsics**. Therefore, for MLLMs to achieve robust and generalizable spatial intelligence, they must be made explicitly *camera-aware*.

## 4 CAMERA-AWARE MLLM FRAMEWORK

We now present our proposed *Camera-Aware MLLM Framework*, designed to explicitly resolve the inherent geometric ambiguity of RGB-only MLLMs for spatial reasoning. A naive solution might be to adapt canonicalization strategies from Metric3D (Yin et al., 2023), which resample all images to a shared virtual camera. However, this is impractical for MLLMs, as it is both computationally prohibitive (causing lots of invalid visual tokens) and relies on precise camera intrinsics that are rarely available for large-scale datasets. Therefore, our framework pursues a more scalable and flexible strategy: instead of transforming the visual data, we condition them directly on camera parameters and geometric priors.

### 4.1 ARCHITECTURE OVERVIEW

As shown in Fig. 4 (a), the model processes text inputs through a standard text encoder. For visual inputs (images or videos), each frame is encoded by a *Geometry-Aware Visual Encoder*, which enriches visual tokens with both raw appearance and geometric context derived from camera intrinsics.

These tokens are then projected into the LLM, following the standard MLLM paradigm (Liu et al., 2023; Bai et al., 2025) of joint multimodal reasoning. The core design lies in Geometry-Aware Visual Encoder (illustrated in Fig. 4 (b)), where we encode camera intrinsics and geometric priors into visual tokens, described next.

### 4.2 CAMERA RAY EMBEDDING

Consistent with standard MLLM architectures, an input image is first processed by a 2D visual encoder (*e.g.*, a Vision Transformer) to produce a grid of visual tokens, denoted as $F_{\text{vis}} \in \mathbb{R}^{H \times W \times D}$.

While these tokens capture rich semantic and appearance information, they are geometrically ambiguous, lacking inherent knowledge of their position within the camera's viewing frustum.

To resolve this ambiguity, we introduce a *Dense Camera Ray Embedding* that explicitly conditions each visual token on its corresponding line-of-sight, derived from the camera's intrinsics. With given intrinsics $(f_x, f_y, c_x, c_y)$, we compute the normalized direction components for each token at grid position $(i, j)$, which corresponds to an image coordinate $(u_{ij}, v_{ij})$: $R_x[i,j] = \frac{u_{ij}-c_x}{f_x}$, and $R_y[i,j] = \frac{v_{ij}-c_y}{f_y}$. We also include the global focal length values, $f_x$ and $f_y$, as part of the embedding for every token. They are then encoded using a sinusoidal embedding layer (Vaswani et al., 2017), generating a dense camera embedding $E_{\text{cam}} \in \mathbb{R}^{H \times W \times D}$, which is then fused with the visual features $F_{\text{vis}}$ via element-wise addition. This straightforward mechanism ensures that each visual token is not just a descriptor of local semantics but is also grounded in its geometric context, making it explicitly aware of its specific line-of-sight into the 3D world.

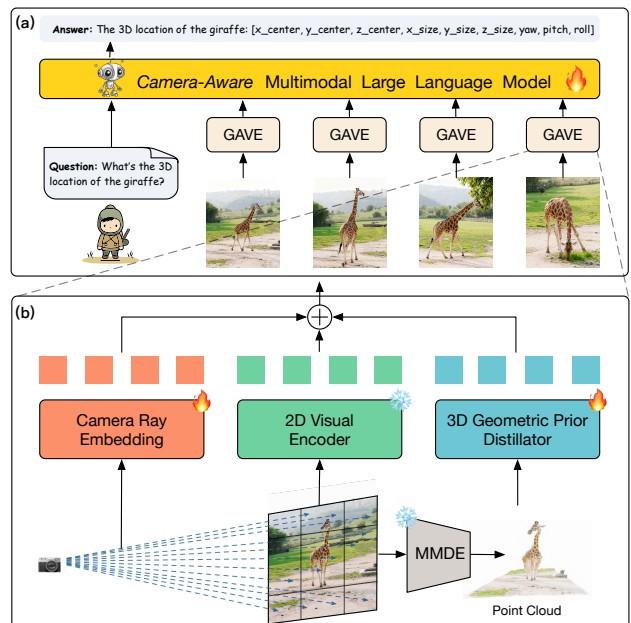

Figure 4: The proposed Camera-Aware MLLM Framework. (a) The overview of the architecture, where (b) Geometry-Aware Visual Encoder (GAVE) injects camera-awareness and 3D geometric priors into the MLLM.

### 4.3 CAMERA-AWARE GEOMETRIC AUGMENTATION

Even with explicit intrinsic conditioning, effective learning requires exposure to diverse cameras. However, most 3D datasets are captured with limited sensor setups, resulting in narrow distributions of intrinsics. To address this limitation, we propose a *Camera-Aware Geometric Augmentation* strategy, as shown in Fig. 5. During training, we synthetically perturb intrinsics by:

- **Scaling:** resizing the image by factor $s$, updating intrinsics as $(f_x, f_y, c_x, c_y) \mapsto (sf_x, sf_y, sc_x, sc_y)$;
- **Shifting:** translating the principal point $(c_x, c_y)$ to simulate off-center projections;

Importantly, both the image and its intrinsics are updated consistently, ensuring geometric correctness. This forces the model to disentangle scene content from camera geometry, equipping it with robustness against cross-camera distribution shifts.

### 4.4 GEOMETRIC PRIOR DISTILLATION

While direct 3D annotations for MLLM training are scarce, large-scale RGB-depth pairs are abundant. To leverage this resource, we distill rich geometric priors from a state-of-the-art Monocular Metric Depth Estimation (MMDE) model, UniDepth v2 (Piccinelli et al., 2025), which was pre-trained on over 10M RGB-depth pairs across diverse cameras.

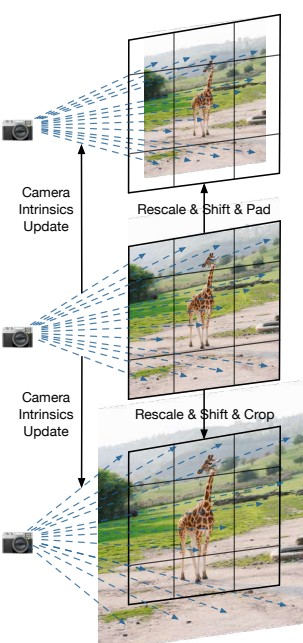

Figure 5: Illustration of camera-aware geometric augmentation.

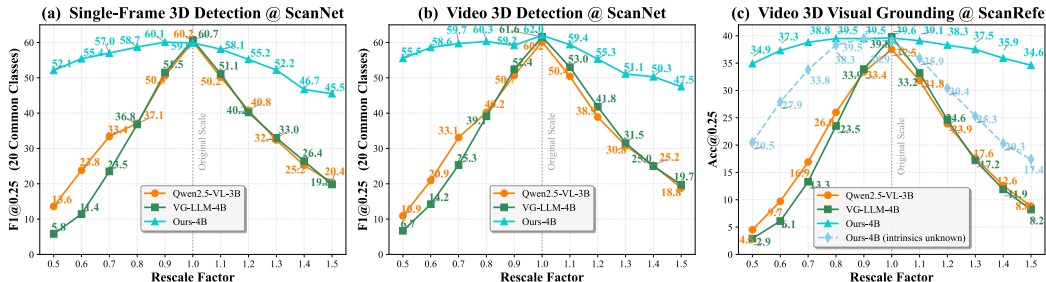

Figure 6: Cross-camera generalization on spatially-grounded tasks. While camera-agnostic MLLMs (Qwen2.5-VL, VG-LLM) fail catastrophically on altered camera geometries by rescaling, our method maintains robust performance, proving its ability to generalize across cameras.

For each training image, we use the frozen UniDepth v2 model to predict a dense 3D point cloud, which is then embedded into a geometric prior embedding $E_{\text{geo}} \in \mathbb{R}^{H \times W \times D}$. $E_{\text{geo}}$ is also added to $F_{\text{vis}}$, further enriching them with explicit 3D structural information.

Notably, geometric prior distillation extends the framework to images with unknown camera parameters, as UniDepth (Piccinelli et al., 2025) is able to estimate intrinsics directly from images. This enables training and evaluation on vast 2D datasets lacking such camera intrinsics annotations.

## 5 EXPERIMENTS

To validate our Camera-Aware MLLM framework and its core claim—that camera-awareness is essential for generalizable spatial reasoning—we conduct three targeted evaluations. We: *(i)* assess the model's cross-camera generalization capacity specifically on spatially-grounded tasks; *(ii)* benchmark its performance on established spatial reasoning benchmarks; and *(iii)* perform a detailed ablation study. In our experiments, we adopt VG-LLM (Zheng et al., 2025) as the baseline. More implementation details are provided in Appendix B

### 5.1 EVALUATION ON CROSS-CAMERA SPATIALLY-GROUNDED TASKS

We test our framework's generalization on a suite of spatially-grounded tasks: single-frame 3D object detection, video 3D object detection, and video 3D visual grounding. As general-purpose models like Gemini-2.5 (Comanici et al., 2025) fail to perform 3D localization, we only compare against task-finetuned baselines: Qwen2.5-VL (Bai et al., 2025) and VG-LLM (Zheng et al., 2025). All models are trained on a large, diverse corpus including ScanNet (Dai et al., 2017), ARK-itScenes (Baruch et al., 2021), Matterport3D (Chang et al., 2017), 3RScan (Wald et al., 2019), SUN RGB-D (Song et al., 2015), Objectron (Ahmadyan et al., 2021), ScanRefer (Chen et al., 2020), and Scan2Cap (Chen et al., 2021). Models are trained on the full mixed-source data. At inference, we test on a ScanNet validation set, but synthetically perturb the camera intrinsics by resizing the input images. This directly tests model robustness against the camera geometry shifts discussed in Sec. 3.3, simulating deployment on cameras with different focal lengths.

As illustrated in Figure 6, our Camera-Aware MLLM demonstrates exceptional robustness to variations in camera intrinsics. In all spatially-grounded tasks, while achieving competitive performance on the original, unscaled data, our model maintains highly consistent accuracy as camera parameters are altered via resizing. In stark contrast, the performance of camera-agnostic baseline MLLMs degrades substantially, confirming their overfitting to the training camera's geometry. These experiments unequivocally demonstrate that our camera-aware approach enables true generalization across cameras with diverse parameters, a critical capability that RGB-only methods fundamentally lack.

It is noteworthy that our framework's robustness extends to scenarios where camera intrinsics are unavailable—a common case for images sourced from the internet. This is made possible by our geometric prior distillation, where the pre-trained MMDE estimates intrinsics on the fly. As shown in Fig. 6 (c), our approach consistently and significantly outperforms camera-agnostic baselines, highlighting its superior generalization even when operating on inputs without explicit camera parameters.

Table 2: Comparison of MLLMs' spatial reasoning performance on SPAR-Bench.

| Methods | Avg. | Low | Depth-OC | Depth-OC-MV | Depth-OO | Depth-OO-MV | Dist-OC | Dist-OC-MV | Dist-OO | Dist-OO-MV | Medium | PosMatch | CamMotion | ViewChg | High | Dist-OO | Dist-OO-MV | ObjRel-OC-MV | ObjRel-OO | ObjRel-OO-MV | Spimg-OC | Spimg-OC-MV | Spimg-OO | Spimg-OO-MV |
|---|---|---|---|---|---|---|---|---|---|---|---|---|---|---|---|---|---|---|---|---|---|---|---|---|
| **SPAR-Bench (tiny)** | | | | | | | | | | | | | | | | | | | | | | | | |
| Human Level | 67.27 | 55.31 | 72.75 | 74.25 | 28.75 | 36.25 | 78.25 | 52.25 | 66.5 | 33.50 | 72.32 | 92 | 64 | 60.97 | 76.22 | 80 | 94 | 70 | 92 | 80 | 78 | 82 | 50 | 60 |
| GPT-4o | 36.39 | 29.25 | 53.80 | 45.00 | 15.00 | 13.60 | 37.40 | 34.40 | 23.40 | 24.40 | 24.93 | 30 | 16 | 28.80 | 45.11 | 64 | 64 | 58 | 46 | 46 | 32 | 44 | 30 | 22 |
| Claude-3.7-Sonnet | 21.77 | 25.43 | 41.00 | 45.40 | 11.20 | 12.20 | 42.60 | 19.60 | 26.00 | 5.40 | 7.33 | 16 | 6 | 0.00 | 23.33 | 40 | 48 | 22 | 36 | 14 | 12 | 20 | 6 | 12 |
| Qwen2-VL-72B | 35.62 | 35.28 | 45.40 | 49.80 | 13.80 | 10.00 | 54.60 | 49.40 | 36.80 | 22.40 | 23.39 | 42 | 18 | 10.16 | 40.00 | 60 | 68 | 50 | 38 | 44 | 18 | 28 | 18 | 36 |
| Qwen2.5-VL-72B | 39.40 | 35.35 | 53.20 | 46.80 | 17.80 | 29.00 | 49.60 | 57.40 | 14.40 | 14.60 | 23.05 | 40 | 16 | 13.16 | 48.44 | 74 | 74 | 60 | 56 | 50 | 20 | 34 | 24 | 44 |
| **SPAR-Bench (full)** | | | | | | | | | | | | | | | | | | | | | | | | |
| InternVL2-8B | 33.02 | 26.83 | 25.75 | 30.88 | 20.67 | 20.78 | 39.03 | 36.19 | 19.15 | 22.19 | 36.49 | 63.36 | 28.00 | 18.11 | 37.37 | 44.71 | 54.46 | 42.75 | 37.36 | 26.32 | 34.14 | 31.10 | 20.86 | 24.65 |
| InternVL2.5-8B | 36.28 | 29.46 | 25.78 | 29.31 | 23.79 | 18.76 | 46.82 | 42.68 | 22.62 | 25.89 | 31.88 | 61.32 | 28.00 | 6.32 | 43.80 | 59.71 | 56.85 | 51.75 | 44.23 | 41.55 | 36.56 | 41.57 | 22.52 | 39.50 |
| LLaVA-OV-7B | 31.20 | 21.79 | 30.33 | 26.94 | 18.58 | 13.87 | 10.43 | 13.64 | 31.24 | 29.29 | 26.13 | 38.68 | 30.25 | 9.47 | 40.14 | 56.47 | 55.06 | 37.25 | 48.63 | 38.23 | 30.33 | 33.72 | 26.49 | 35.01 |
| Qwen2.5-VL-7B | 33.07 | 28.75 | 31.33 | 33.66 | 21.99 | 14.97 | 42.88 | 37.73 | 23.83 | 23.64 | 22.97 | 33.33 | 28.75 | 6.83 | 40.27 | 58.24 | 51.49 | 44.75 | 50.00 | 32.13 | 33.87 | 32.85 | 27.15 | 31.93 |
| LLaVA-v1.5-7B | 23.65 | 10.85 | 5.17 | 12.53 | 17.37 | 11.34 | 7.25 | 5.26 | 18.73 | 9.12 | 26.50 | 24.43 | 26.75 | 28.31 | 34.09 | 51.18 | 52.38 | 34.25 | 24.18 | 26.87 | 34.68 | 29.94 | 22.52 | 30.81 |
| LLaVA-v1.6-7B | 13.21 | 8.53 | 12.14 | 0.00 | 20.35 | 0.27 | 10.76 | 0.41 | 24.27 | 0.00 | 4.79 | 6.62 | 7.75 | 0.00 | 20.18 | 51.76 | 7.74 | 6.25 | 32.14 | 6.37 | 39.52 | 10.47 | 21.52 | 5.88 |
| GPT-5 | 37.40 | - | - | - | - | - | - | - | - | - | - | - | - | - | - | - | - | - | - | - | - | - | - | - |
| Gemini-2.5-Pro | 36.30 | - | - | - | - | - | - | - | - | - | - | - | - | - | - | - | - | - | - | - | - | - | - | - |
| VG-LLM-4B | 60.36 | 52.81 | 83.97 | 44.75 | 34.27 | 16.96 | 79.08 | 60.63 | 66.20 | 36.63 | 51.35 | 39.69 | 63.00 | 25.82 | 72.92 | 89.12 | 63.69 | 81.25 | 83.79 | 77.84 | 68.82 | 60.17 | 59.60 | 71.99 |
| SPAR-8B | 63.25 | 65.53 | 81.53 | 79.22 | 38.25 | 35.51 | 78.93 | 79.18 | 68.13 | 63.50 | 63.01 | 78.88 | 73.00 | 37.14 | 61.29 | 86.47 | 79.76 | 64.00 | 69.23 | 59.00 | 47.31 | 50.00 | 42.38 | 53.50 |
| **Ours-4B** | 68.35 | 59.94 | 89.61 | 49.12 | 57.07 | 19.27 | 88.02 | 63.40 | 77.60 | 35.42 | 60.42 | 47.84 | 73.00 | 31.01 | 81.74 | 92.35 | 68.75 | 87.50 | 91.76 | 84.21 | 79.57 | 66.86 | 83.44 | 81.23 |

• Tasks are split into 'low-level', 'medium-level', and 'high-level' tasks based on different complexities, following SPAR-Bench authors.

Table 3: Comparison of MLLMs' spatial reasoning performance on VSI-Bench.

| | Avg. | Obj. Count | Abs. Dist. | Obj. Size | Room Size | Rel. Dist. | Rel. Dir. | Route Plan | Appr. Order |
|---|---|---|---|---|---|---|---|---|---|
| | | Numerical Answer | | | | Multiple-Choice Answer | | | |
| *Proprietary Generalist MLLMs:* | | | | | | | | | |
| GPT-4o | 34.0 | 46.2 | 5.3 | 43.8 | 38.2 | 37.0 | 41.3 | 31.5 | 28.5 |
| GPT-5-Chat-0807-global | 49.1 | 52.4 | 36.6 | 68.0 | 55.5 | 49.9 | 47.4 | 33.0 | 49.8 |
| Gemini-1.5-Flash | 42.1 | 49.8 | 30.8 | 53.5 | 54.4 | 37.7 | 41.0 | 31.5 | 37.8 |
| Gemini-1.5-Pro | 45.4 | 56.2 | 30.9 | 64.1 | 43.6 | 51.3 | 46.3 | 36.0 | 34.6 |
| Gemini-2.5-Pro-0617 | 52.7 | 48.2 | 35.6 | 71.3 | 51.7 | 58.9 | 42.4 | 46.8 | 66.7 |
| *Open-Sourced Generalist MLLMs:* | | | | | | | | | |
| InternVL2-8B | 34.6 | 23.1 | 28.7 | 48.2 | 39.8 | 36.7 | 30.7 | 29.9 | 39.6 |
| InternVL2-40B | 36.0 | 34.9 | 26.9 | 46.5 | 31.8 | 42.1 | 32.2 | 34.0 | 39.6 |
| Qwen2.5VL-3B | 30.6 | 24.3 | 24.7 | 31.7 | 22.6 | 38.3 | 41.6 | 26.3 | 21.2 |
| Qwen2.5VL-72B | 37.0 | 25.1 | 29.3 | 54.5 | 38.8 | 38.2 | 37.0 | 34.0 | 28.9 |
| LongVILA-8B | 21.6 | 29.1 | 9.1 | 16.7 | 0.0 | 29.6 | 30.7 | 32.5 | 25.5 |
| VILA-1.5-40B | 31.2 | 22.4 | 24.8 | 48.7 | 22.7 | 40.5 | 25.7 | 31.5 | 32.9 |
| LongVA-7B | 29.2 | 38.0 | 16.6 | 38.9 | 22.2 | 33.1 | 43.3 | 25.4 | 15.7 |
| LLaVA-NeXT-Video-72B | 40.9 | 48.9 | 22.8 | 57.4 | 35.3 | 42.4 | 36.7 | 35.0 | 48.6 |
| LLaVA-OneVision-72B | 40.2 | 43.5 | 23.9 | 57.6 | 37.5 | 42.5 | 39.9 | 32.5 | 44.6 |
| VideoLLaMA3-7B | 35.8 | 41.9 | 23.5 | 42.2 | 27.1 | 39.4 | - | 32.0 | 31.4 |
| *MLLMs for Spatial Intelligence:* | | | | | | | | | |
| SAT-LLaVA-Video-7B | - | - | - | - | 47.3 | 41.1 | 37.1 | 36.1 | 40.4 |
| SPAR-8B | 41.1 | - | - | - | - | - | - | - | - |
| RynnEC-7B | 45.8 | 58.5 | 25.4 | 54.9 | 42.7 | 44.2 | - | 38.7 | 30.5 |
| VG-LLM-4B | 47.3 | 66.0 | 37.8 | 55.2 | 59.2 | 44.6 | 45.6 | 33.5 | 36.4 |
| Ours-4B | 46.8 | 71.3 | 39.3 | 50.4 | 65.9 | 50.7 | 45.5 | 31.4 | 20.2 |

## 5.2 EVALUATION ON GENERAL SPATIAL REASONING BENCHMARKS

A key advantage of our framework is its remarkable flexibility in handling diverse visual inputs. Our model can seamlessly process both data with known camera intrinsics, common in embodied AI, and standard RGB images or videos where such information is absent. When intrinsics are absent, we utilize the MMDE to estimate the intrinsics, enabling us to train a single, powerful generalist model for spatial reasoning on a large-scale, heterogeneous corpus. To this end, we combine the LLaVA-Hound split of LLaVA-Video-178k (Zhang et al., 2024b), SPAR (Zhang et al., 2025), and our own curated collection of spatially-grounded tasks to train a generalist model on spatial reasoning.

Table 4: Comparison of model performance on various spatial understanding datasets.

| Model | CV-Bench-3D | | | BLINK-Spatial | | | |
|---|---|---|---|---|---|---|---|
| | Avg. | Depth | Dist. | Avg. | Depth | Spatial Rel. | Multi. View |
| GPT-4V | 69.1 | - | - | 62.8 | 60.0 | 72.7 | 55.6 |
| GPT-4o | 83.0 | - | - | 67.6 | 74.2 | 69.2 | 59.4 |
| Gemini-2.5-Pro | 90.8 | 91.0 | 90.7 | 71.6 | 87.9 | 91.6 | 35.3 |
| Qwen2.5-VL-3B | 69.5 | 74.5 | 66.2 | 64.7 | 68.5 | 81.1 | 44.4 |
| Qwen2.5-VL-7B | 77.3 | 84.5 | 76.5 | 73.8 | 79.0 | 86.7 | 55.6 |
| Qwen2.5-VL-32B | 79.1 | 83.0 | 83.2 | 66.8 | 69.4 | 86.0 | 45.1 |
| Qwen2.5-VL-72B | 81.0 | 88.7 | 81.0 | 68.6 | 79.8 | 85.3 | 40.6 |
| SAT-LLaVA-Video-7B | 78.4 | - | - | 62.6 | 66.1 | 73.4 | 48.1 |
| SPAR-8B | 89.1 | - | - | - | - | - | - |
| VG-LLM-4B | 91.3 | - | - | 68.4 | 79.8 | 71.3 | 54.1 |
| Ours-4B | 90.7 | 89.8 | 91.5 | 77.0 | 77.4 | 66.4 | 87.2 |

We evaluate this generalist model on a range of established spatial reasoning benchmarks against state-of-the-art methods. As shown in Table 2, on SPAR-Bench, which provides precise camera parameters, our model achieves the highest performance, directly validating the effectiveness of our

camera-aware design in a controlled setting. As shown in Table 3 and Table 4, even on general spatial reasoning benchmarks designed for RGB-only methods, where intrinsics are not provided, our method still achieves the state-of-the-art performance. These results demonstrate that our camera-aware approach is not a narrow fix, but a foundational principle that enables robust and generalizable spatial intelligence when scaled with diverse data.

### 5.3 ABLATION STUDY

To dissect the contributions of our framework, we conducted an ablation study on the cross-camera generalization task of single-frame 3D object detection. We evaluated performance on a synthetically "zoomed-in" test set—created by rescaling and centrally cropping images—to simulate an out-of-distribution camera with altered intrinsics. We adopt VG-LLM (Zheng et al., 2025) as our baseline to quantify the improvements.

Table 5: Ablation study on the components of our Camera-Aware MLLM framework. Performance measured on ScanNet-val x1.2 to test cross-camera generalization.

| Components | | | 20 Common Classes | | |
|---|---|---|---|---|---|
| Ray Emb. | Geom. Aug. | Prior Dist. | $P_{0.25}$ | $R_{0.25}$ | $\boldsymbol{F1_{0.25}}$ |
| - | - | - | 40.8 | 37.7 | 39.1 |
| ✓ | - | - | 42.9 | 39.7 | 41.2 |
| - | ✓ | - | 43.8 | 40.4 | 42.0 |
| - | - | ✓ | 45.7 | 41.1 | 43.1 |
| ✓ | - | ✓ | 46.2 | 42.8 | 44.3 |
| ✓ | ✓ | ✓ | **54.6** | **50.1** | **52.1** |

Our ablation in Table 5 demonstrates that neither a camera-aware architecture nor diverse training data is sufficient on its own. While adopting a camera-aware architecture or employing geometric augmentation alone provides moderate gains, it is their synergy that unlocks substantial improvements in generalization. This combination forces the model to internalize geometric principles, proving that both architectural camera awareness and data-level diversity are indispensable for achieving robust spatial intelligence in MLLMs.

### 5.4 VISUALIZATION RESULTS

Fig. 7 qualitatively compares the 3D detection performance of our proposed Camera-Aware MLLM against camera-agnostic baselines trained with identical data. The visualizations reveal a significant gap in localization accuracy on both zoomed-in images from ScanNet and in-the-wild images from the unseen TUM-RGBD dataset (Sturm et al., 2012). This demonstrates that by both incorporating camera parameters and being exposed to diverse camera setups via our camera-aware geometric augmentation, our framework learns fundamental geometric principles. This mitigates the ambiguities that hinder baselines, leading to substantially more accurate and generalizable spatial understanding and reasoning.

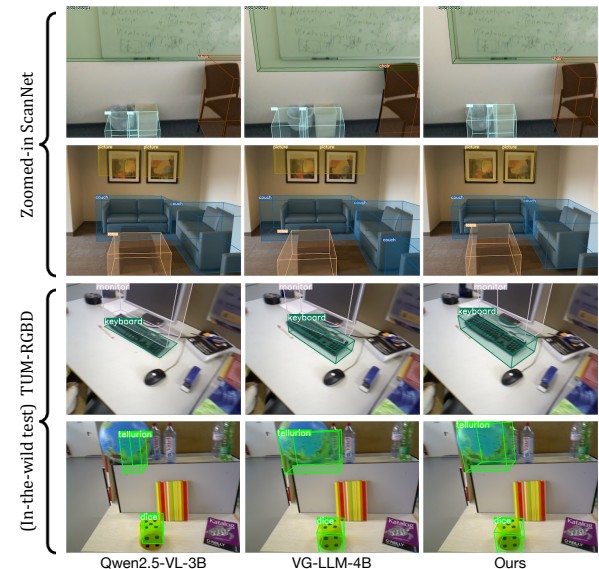

Figure 7: Qualitative comparison of 3D object detection results. Our camera-aware MLLM can robustly generalize to new camera setups.

## 6 CONCLUSION

In this work, we have demonstrated that camera-agnostic MLLMs, despite their apparent success, are fundamentally flawed for spatial reasoning tasks due to an irresolvable geometric ambiguity. By ignoring camera intrinsics, these models fail to generalize, instead overfitting to the specific camera properties of their training data. Our proposed Camera-Aware MLLM framework mitigates this issue by injecting camera parameters, employing a geometric augmentation strategy, and distilling 3D priors. We argue for a paradigm shift: to build truly generalizable MLLMs that understand our 3D world, we must move beyond processing solely pixels to reasoning about the geometric principles that create them.

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

## A    DETAILED PROOF OF GEOMETRIC AMBIGUITY

We formalize the ambiguity that arises when camera intrinsics are unknown. Unless otherwise stated, we assume an ideal pinhole camera and ignore lens distortion and skew; including them does not change the conclusions.

**Camera projection model.**    Let a 3D point in the world frame be $\mathbf{P}_w \in \mathbb{R}^3$. The standard projection is

$$s \begin{bmatrix} u \\ v \\ 1 \end{bmatrix} = \mathbf{K} \left[ \mathbf{R} \mid \mathbf{t} \right] \begin{bmatrix} \mathbf{P}_w \\ 1 \end{bmatrix}, \qquad \mathbf{K} = \begin{bmatrix} f_x & 0 & c_x \\ 0 & f_y & c_y \\ 0 & 0 & 1 \end{bmatrix}, \tag{3}$$

where $\mathbf{R} \in \mathrm{SO}(3)$ and $\mathbf{t} \in \mathbb{R}^3$ are extrinsics, and $s \neq 0$ is the projective scale. In the camera frame, letting $\mathbf{P}_c = \mathbf{R}\mathbf{P}_w + \mathbf{t} = (X, Y, Z)^\top$, we have

$$\mathbf{K}\mathbf{P}_c = \begin{bmatrix} f_x X + c_x Z \\ f_y Y + c_y Z \\ Z \end{bmatrix} \quad \Rightarrow \quad s = Z, \quad u = \frac{f_x X}{Z} + c_x, \quad v = \frac{f_y Y}{Z} + c_y. \tag{4}$$

**Projected extent under a fronto-parallel configuration.**    Consider a vertical, fronto-parallel segment of physical height $H$ (in meters) centered on the optical axis: its top and bottom endpoints in the camera frame are $\mathbf{P}_{\text{top}} = (0,\ H/2,\ Z)^\top$ and $\mathbf{P}_{\text{bottom}} = (0,\ -H/2,\ Z)^\top$. From equation 4,

$$v_{\text{top}} = \frac{f_y(H/2)}{Z} + c_y, \qquad\qquad v_{\text{bottom}} = \frac{f_y(-H/2)}{Z} + c_y, \tag{5}$$

so the image height in pixels is

$$h_{\text{proj}} = v_{\text{top}} - v_{\text{bottom}} = \frac{f_y H}{Z}. \tag{6}$$

Analogously, for a horizontal extent $W$ we have $w_{\text{proj}} = \frac{f_x W}{Z}$.

**Coupled-scaling invariance (focal–depth & size–depth).**    From equation 6, the projected height is $h_{\text{proj}} = \frac{f_y H}{Z}$. It is invariant under the coupled rescaling

$$(f_y,\ H,\ Z) \ \mapsto\ (\alpha f_y,\ \beta H,\ \alpha\beta Z) \quad (\alpha, \beta > 0),$$

since

$$h'_{\text{proj}} = \frac{(\alpha f_y)(\beta H)}{\alpha\beta Z} = h_{\text{proj}}.$$

Two canonical cases:

- **Focal–depth tradeoff:** set $\beta = 1$. Increasing focal length by $\lambda$ while moving the object to depth $\lambda Z$ leaves $h_{\text{proj}}$ unchanged (optical/digital zoom vs. depth).
- **Size–depth tradeoff:** set $\alpha = 1$. Scaling physical size and depth by the same factor $\lambda$ leaves $h_{\text{proj}}$ unchanged (small-near vs. large-far).

Thus, from a single RGB image, a nearby small object can be observationally indistinguishable (in projected height) from a larger distant one, and a change in depth can be indistinguishable from a change in focal length.

**Image resampling as intrinsic transformation.**    Any pre-processing that rescales pixel coordinates by $(\sigma_x, \sigma_y)$ (without cropping) is equivalent to

$$(u', v', 1)^\top = \mathrm{diag}(\sigma_x, \sigma_y, 1)\,(u, v, 1)^\top \quad \Longleftrightarrow \quad \mathbf{K}' = \mathrm{diag}(\sigma_x, \sigma_y, 1)\,\mathbf{K}.$$

A subsequent crop with top-left offset $(\Delta u, \Delta v)$ (measured after rescaling) shifts the principal point as $c'_x = \sigma_x c_x - \Delta u$, $c'_y = \sigma_y c_y - \Delta v$.

**Per-pixel rays and the role of intrinsics.** The back-projected ray for pixel $(u, v)$ in camera coordinates is, up to scale,

$$\mathbf{d} \propto \mathbf{K}^{-1} \begin{bmatrix} u \\ v \\ 1 \end{bmatrix} = \begin{bmatrix} (u - c_x)/f_x \\ (v - c_y)/f_y \\ 1 \end{bmatrix}. \tag{7}$$

Hence the entire ray field $\{\mathbf{d}(u, v)\}$ depends on $(f_x, f_y, c_x, c_y)$. If intrinsics are unknown, two images that *appear* similar can correspond to different ray bundles, encouraging camera-specific shortcuts rather than geometry that transfers across cameras.

**Scope and assumptions.** The invariance in equation 6 is derived for a fronto-parallel segment and concerns the *observable projected extent* (*e.g.*, $h_{\text{proj}}$). For non-fronto-parallel shapes or general scenes, pixel-wise equality need not hold under the above transformations; nevertheless, the *metric identifiability* issue at the level of absolute size and depth persists: with monocular RGB and unknown intrinsics, the mapping in equation 4 is homogeneous in $(X, Y, Z)$, so metric depth and size cannot be uniquely recovered without additional metric priors, multiple views, or known intrinsics. Providing $\mathbf{K}$ (or its sufficient statistics, such as per-pixel ray directions) is therefore a necessary information channel to resolve the focal–depth ambiguity and enable cross-camera generalization.

# B  IMPLEMENTATION DETAILS

Our work builds upon the VG-LLM-4B (Zheng et al., 2025) framework, which we adopt as our baseline. The VG-LLM-4B model integrates Qwen2.5-VL-3B (Bai et al., 2025) with VGGT-1B (Wang et al., 2025) for 3D geometry feature extraction. To further enhance the model's geometric understanding ability, we employ UniDepth v2 (Piccinelli et al., 2025) to distill and inject rich 3D geometric priors into the framework. We train the model for one epoch using the Adam optimizer with a warmup ratio of 0.03. The learning rate is gradually increased to 1e-5 and subsequently decayed. The batch size is set to 64 in total. During training, the visual encoder of Qwen2.5-VL, the 3D geometry encoder (VGGT), and UniDepth v2 are frozen, while the MLLM, camera ray embedding, and the 3D geometric prior distillator remain trainable. All experiments are conducted on 8 H100 80G GPUs.

We adopt a mixed dataset for training. For spatially-grounded tasks (Sec 5.1), our model is trained on ScanNet (Dai et al., 2017), ARKitScenes (Baruch et al., 2021), Matterport3D (Chang et al., 2017), 3RScan (Wald et al., 2019), SUN RGB-D (Song et al., 2015), Objectron (Ahmadyan et al., 2021), ScanRefer (Chen et al., 2020), and Scan2Cap (Chen et al., 2021). For spatial reasoning tasks (Sec 5.2), our model is trained on the LLaVA-Hound split of LLaVA-Video-178k (Zhang et al., 2024b), SPAR (Zhang et al., 2025), as well as the aforementioned curated collection of spatially-grounded datasets. For ablation studies 5.3, our model is trained on the task of single-frame 3D object detection based on the ScanNet (Dai et al., 2017).

Table 6: The prompt for 3D video object detection.

$\mathbf{X}_{\text{User}}$: `<image><image><image><image>`···
Detect the 3D locations for the following categories: curtain, bathtub, towel, doorframe, and bar.
Output a json list where each entry contains the object name in "label" and its 3D bounding box in "box_3d".
The 3D bounding box format should be [x_center, y_center, z_center, x_size, y_size, z_size, yaw, pitch, rolll].
$\mathbf{X}_{\text{Agent}}$:
```json
[
    {"label": "curtain", "bbox_3d": [-0.5, -0.0, 0.7, 0.9, 0.4, 2.0, -2.5, 1.1, -2.9]},
    {"label": "bathtub", "bbox_3d": [-0.3, -0.6, -1.2, 2.6, 0.8, 0.7, -2.0, 1.0, -2.7]},
    ...
]
```

To prevent hallucinations in 3D video object detection (such as the inclusion of categories that are absent from the scene), we explicitly provide the categories in the input. An example prompt for 3D video object detection is presented in Table 6.

## C    LLM USAGE DISCLOSURE

Large Language Models (LLMs) were utilized as a general-purpose assist tool during the preparation of this manuscript. Specifically, LLMs aided in polishing the language and improving the overall readability of the paper, assisting with various writing tasks, and also providing support for coding-related aspects of the research.

The authors confirm that any LLM-generated content was based on the authors' original ideas and experiment results, thoroughly reviewed and verified by the authors, and accurately reflects the intended meaning. LLMs were not used to autonomously complete any part of the paper.

