# OpenReview forum: "On the Generalization Capacities of MLLMs for Spatial Intelligence"
_ICLR.cc/2026/Conference — ICLR 2026 Oral_

### Official Review · Reviewer_z9kB · 2025-10-29

**Soundness:** 3
**Presentation:** 3
**Contribution:** 2
**Rating:** 6
**Confidence:** 3

**Summary:**

This paper investigates the limitations of "RGB-only" multimodal large language models in spatial reasoning tasks, particularly their lack of generalization across camera viewpoints. The authors attribute this to the absence of camera intrinsics, which they argue introduces geometric ambiguity and leads to overfitting. To address this, the paper introduces a Camera-Aware MLLM framework that incorporates dense camera embeddings, camera-aware data augmentation, and geometric prior distillation from a 3D vision model.

**Strengths:**

(1) This paper is well-motivated. It identifies a fundamental limitation in existing MLLMs for spatial tasks, based on both theory and empirical analysis. What’s more, the cross-camera generalization experiments are particularly compelling and clearly demonstrate the value of the proposed method.

(2) The proposed Camera-Aware MLLM framework is evaluated through a series of experiments. The results show consistent improvements over camera-agnostic baselines, suggesting that incorporating camera parameters can enhance the robustness of spatial MLLMs.

**Weaknesses:**

(1) Limited discussion on real-world applicability. The paper would benefit from a discussion about the real-world scenarios where camera intrinsics may be unknown, unreliable, or hard to estimate depth.

(2) The ablation study in Table 5 lacks the “Only Prior Distillation” setting, which I consider important for understanding the individual contribution of geometric prior supervision.

(3) It remains unclear whether the improvements stem from the proposed model itself or simply from the availability of additional 3D geometric information. It could be clarified by comparing with methods that directly utilize depth maps or incorporate depth-aware features into the model.

**Questions:**

See Weaknesses

---

> ### Author Response · Authors · 2025-11-24
>
> Thank you for the positive assessment of our work, particularly for recognizing our motivation as "well-founded" and our cross-camera generalization experiments as "compelling." We appreciate your constructive feedback, which has helped us strengthen our paper. Below, we address the specific concerns raised.
>
> &nbsp;
>
>
> --------
>
> &nbsp;
>
> ### Real-World Applicability (Unknown or Unreliable Camera Intrinsics)
>
> We fully agree that discussing real-world scenarios with varying intrinsic availability is crucial. In our experiments, we have extensively validated our model in both "known" and "unknown" intrinsic settings.
>
> **Performance with Unknown Intrinsics on Spatially-grounded tasks:** In **Fig.6(c)**, we explicitly tested our model in settings where intrinsics are unknown. In this case, the model relies on estimated intrinsics. As shown in Fig. 6(c) (experiment results sorted in the table below), while there is a natural performance gap compared to using accurate intrinsics due to estimation noise, our method (w/o camera intrinsics) still consistently outperforms the camera-agnostic baselines.
>
> | Rescale Factor | Qwen2.5-VL-3B | VG-LLM-4B | Ours-4B | Ours-4B (intrinsics unknown) |
> | :---: | :---: | :---: | :---: | :---: |
> | **0.5** | 4.5 | 2.9 | 34.9 | 20.5 |
> | **0.6** | 9.7 | 6.1 | 37.3 | 27.9 |
> | **0.7** | 16.9 | 13.3 | 38.8 | 33.8 |
> | **0.8** | 26.0 | 23.5 | 39.5 | 38.3 |
> | **0.9** | 33.4 | 33.9 | 39.5 | 39.5 |
> | **1.0** | 37.5 | 39.8 | 39.6 | 38.9 |
> | **1.1** | 31.8 | 33.2 | 39.1 | 35.9 |
> | **1.2** | 23.9 | 24.6 | 38.3 | 30.4 |
> | **1.3** | 17.6 | 17.2 | 37.5 | 25.3 |
> | **1.4** | 12.6 | 11.9 | 35.9 | 20.3 |
> | **1.5** | 8.8 | 8.2 | 34.6 | 17.4 |
>
> &nbsp;
>
>
> **SPAR-Bench (with camera intrinsics):** This benchmark (Table 2 in the paper) explicitly provides accurate camera parameters, simulating the primary setting for real-world applications (e.g., robots, autonomous driving, surveillance, motion capture). In this setting, where our framework can fully utilize exact intrinsics, our model (68.35) achieves a massive improvement. We outperform the much larger Qwen2.5-VL-72B (39.40) by nearly 30 points, outperform the previous SOTA (63.25), and even surpass the Human Level (67.27). This proves our method is decisively superior when camera parameters are available.
>
> **VSI-Bench, CV-Bench, BLINK (no camera intrinsic):** On benchmarks like VSI-Bench, CV-Bench, and BLINK (Tables 3 and 4 in our paper) where intrinsics are absent, our model relies on estimated intrinsics. Consistent with our analysis in Fig. 6(c), while the reliance on estimated intrinsics introduces noise that narrows the performance gap, our method still achieves clear performance gains compared to counterparts.
>
> These extensive experiments validate our method's robustness and effectiveness across diverse settings, ranging from ideal robotics scenarios to unconstrained internet videos.
>
> &nbsp;
>
> ### Ablation Study
>
> Thank you for this valuable suggestion. As requested, we have updated Table 5 to include the "Only Prior Distillation" setting (highlighted in red) to isolate the contribution of geometric priors.
>
> **Updated Table 5: Ablation study on the components of our Camera-Aware MLLM framework. Performance measured on ScanNet-val x1.2 to test cross-camera generalization.**
> | Ray Emb. | Geom. Aug. | Prior Dist. | $P_{0.25}$ | $R_{0.25}$ | **$F1_{0.25}$** |
> | :---: | :---: | :---: | :---: | :---: | :---: |
> | - | - | - | 40.8 | 37.7 | 39.1 |
> | ✓ | - | - | 42.9 | 39.7 | 41.2 |
> | - | ✓ | - | 43.8 | 40.4 | 42.0 |
> | - | - | ✓ | 45.7 | 41.1 | 43.1 |
> | ✓ | - | ✓ | 46.2 | 42.8 | 44.3 |
> | ✓ | ✓ | ✓ | **54.6** | **50.1** | **52.1** |
>
> &nbsp;
>
>
> ### Source of Improvements
>
> Thanks for the insightful question. The results in the updated Table 5 provide a clearer answer:
>
> If the improvement stemmed solely from the distilled 3D geometric information, the "Only Prior Distillation" setting would have achieved performance comparable to our full model. However, the gain is limited (43.1), leaving a significant gap compared to our full method (52.1).
>
> Substantial improvement is only achieved when all modules within our Camera-Aware MLLM framework work in synergy. By simultaneously leveraging data diversity and explicit encoding of camera parameters, our framework forces the model to internalize fundamental geometric principles, rather than merely attending to auxiliary features.
>
> Furthermore, regarding the suggestion to compare with methods that directly utilize depth maps, we respectfully note that such a comparison would be unfair. Depth-based methods typically rely on ground-truth depth captured by specialized active sensors (e.g., LiDAR, ToF). In contrast, our approach operates in an RGB-only paradigm, making it applicable to a much wider range of real-world scenarios where depth sensors are unavailable.
> &nbsp;
>
>
> -----
>
> &nbsp;
>
>
> We hope these responses address your concerns. We would really appreciate it if you could consider supporting our work with an increased rating.

---

> > ### Comment · Reviewer_z9kB · 2025-11-27
> >
> > Thanks for your rebuttal. The response has addressed my concern about scenarios where camera intrinsics may be unknown or unreliable. The proposed method appears capable of handling such challenging settings effectively. I still have a few suggestions or questions for further improving the manuscript:
> >
> > 1. Missing VG-LLM baseline in Table 2. As an important baseline, including VG-LLM’s results would better demonstrate the model’s improvements over standard spatial reasoning benchmarks.
> > 2. 3D visual grounding performance. The model should be evaluated on 3D benchmarks such as ScanRefer or SQA3D. These tasks usually require reasoning over explicit 3D representations rather than only single-view or multi-view images (e.g., VSI or SPAR). A comparison with 3D-enhanced methods would help clarify that the performance gains do not arise from additional 3D geometric information.
> > 3. VG LLM score inconsistency. Some VG LLM scores reported in the manuscript appear inconsistent with the latest arXiv version. For example, Table 3 lists a VSI-Bench score of 46.1, whereas the most recent VG LLM paper reports 47.3. It should be corrected in the final revision.

---

> ### Author Response · Authors · 2025-11-30
>
> We thank the reviewer for the prompt response and for acknowledging that our previous rebuttal effectively addressed the concerns regarding "unknown/unreliable intrinsics."
>
> Below, we address your remaining suggestions.
>
> &nbsp;
>
> -----
>
> &nbsp;
>
>
> ### 1. Missing VG-LLM Baseline in Table 2
>
> Thank you for this suggestion. We have added **VG-LLM** to **Table 2** (SPAR-Bench) in the revised manuscript to provide a more comprehensive comparison.
> * **VG-LLM-4B Score:** 60.36 (Avg)
> * **Ours-4B Score:** **68.35** (Avg)
> * _(Please see the updated paper for detailed performance.)_
>
> Our method significantly outperforms the VG-LLM baseline, demonstrating the clear advantage of explicit camera modeling on spatial reasoning benchmarks.
>
> &nbsp;
>
> &nbsp;
>
>
>
>
>
>
>
>
> ### 2. 3D Visual Grounding Performance
>
> #### **ScanRefer:**
>
> We clarify that **we had already evaluated our performance on the ScanRefer benchmark** in our initial submission (see **Fig. 6(c)**).
>
> Per your suggestion to compare our method with 3D-enhanced methods, we provide the comparison in Table R3 below.
>
> **Table R3: Comparison on ScanRefer (Acc@0.25).**
> | Method | Input Modality | Acc@0.25 |
> | :--- | :---: | :---: |
> | **3D Scene Input Methods** | *(Uses GT 3D Point Clouds)* | |
> | 3D-LLM | 3D Scene | 30.3 |
> | Chat-3D v2 | 3D Scene | 35.9 |
> | LLaVA-3D | 3D Scene + RGB Video | 54.1 |
> | Video-3D LLM | 3D Scene + RGB Video | 58.1 |
> | :--- | :---: | :---: |
> | **RGB Input Methods** | *(Uses only 2D Images/Videos)* | |
> | **Ours-4B** | **RGB Video** | **39.6** |
>
> We respectfully emphasize that a direct comparison between **RGB-based methods** (like ours) and **3D-input methods** (like LLaVA-3D) involves a fundamental difference in input modality. 3D-input methods take **complete, ground-truth 3D scene representations (point clouds)** as input, thereby bypassing the camera ambiguity problem entirely. In contrast, our method operates in the **RGB-only paradigm**, where the model must infer geometry from 2D pixels without access to ground-truth 3D structures.
>
> While our performance naturally trails behind methods that have "oracle" access to 3D point clouds, our _Ablation Study (Table 5)_ explicitly validates the source of our gains. The results show that improvements are not solely driven by the distilled geometric priors; rather, the Ray Embedding and Geometric Augmentation provide complementary and significant boosts (+9.0 F1 combined). This confirms that our performance stems from the holistic camera-aware design and the synergy of its modules, not merely from the injection of estimated depth information.
>
> &nbsp;
>
> #### **SQA3D:**
>
> As suggested, we further evaluated our model on **SQA3D**, a benchmark requiring reasoning over explicit 3D representations. We finetuned both our model and the baseline on the SQA3D training set.
>
> **Table R4: Comparison on SQA3D.**
> | Method | Input Modality | Accuracy |
> | :--- | :---: | :---: |
> | VG-LLM-4B (Baseline) | RGB Video | 57.91 |
> | **Ours-4B** | **RGB Video** | **60.39** |
>
> **Analysis:**
> Our method outperforms the strong baseline VG-LLM by **+2.48%**. This further confirms that our camera-aware framework effectively enhances spatial reasoning capabilities in complex 3D environments, even when operating with RGB-only inputs.
>
> &nbsp;
>
> &nbsp;
>
>
>
>
>
>
>
> ### 3. VG-LLM Score Inconsistency
>
> We appreciate the reviewer's meticulous attention to detail.
>
> The discrepancy arises because the authors of VG-LLM refactored their codebase (optimizing data processing, learning rates, and training recipes) and then released an updated arXiv version on **October 22, 2025 (after the ICLR 2026 submission deadline)**. The scores in our initial submission were based on the version available before that update.
>
> To ensure our paper is consistent with the latest state of the field, we have updated Table 3 in our revised manuscript to match the latest reported numbers from the VG-LLM arXiv (v3) update (e.g., VSI-Bench: 47.3).
>
> &nbsp;
>
> ---
>
> &nbsp;
>
> We sincerely appreciate your constructive feedback. Your meticulous attention to detail clearly reflects your deep expertise in this domain.
>
> We hope that our new experiments and clarifications regarding the baselines and input modalities have fully resolved your remaining concerns.
>
> &nbsp;

---

### Official Review · Reviewer_VBue · 2025-11-01

**Soundness:** 2
**Presentation:** 2
**Contribution:** 2
**Rating:** 4
**Confidence:** 4

**Summary:**

This paper identifies a fundamental limitation in RGB-only Multimodal Large Language Models (MLLMs) for spatial reasoning tasks: by ignoring camera intrinsic parameters, these models cannot disambiguate geometric properties from camera perspectives, leading to poor cross-camera generalization. The authors propose a Camera-Aware MLLM framework that addresses this issue through three key novelties: (i) dense camera ray embeddings that condition each visual token on its corresponding camera geometry, (ii) camera-aware data augmentation that synthetically varies camera parameters during training, and (iii) geometric prior distillation from a pre-trained monocular depth estimation model (UniDepth v2).

**Strengths:**

1. The paper demonstrates that camera-agnostic MLLMs fail to generalize across different camera parameters, highlighting a critical but overlooked limitation in current spatial reasoning approaches.
2. The proposed framework combines three complementary components to resolve the identified ambiguity, with experimental results showing improvements in cross-camera generalization on multiple spatially-grounded tasks.

**Weaknesses:**

1. Table 5 does not include "Prior Distillation only" or "Geom Aug + Prior Dist (w/o Ray Emb)", making it hard to isolate the contribution of camera ray embeddings. Given that UniDepth v2 is pretrained on 10M+ RGB-depth pairs, the geometric prior distillation likely contributes the majority of improvements, but this is never quantified.

2. Outdated and incomplete baselines:
* Table 2/3/4 use Gemini-1.5-Flash/Pro (Feb 2024) instead of Gemini-2.5 or later versions
* Table 4 is missing Qwen2.5-VL-7B and Qwen2.5-VL-72B, which are natural baselines given the authors' use of Qwen2.5-VL-3B
* No comparison with Claude 4.5 Sonnet, GPT-5 or other recent strong MLLMs

3. Table 1 and Figure 6 only test uniform image rescaling (0.8×, 1.2×), which is an extremely limited simulation of camera variation. Real cross-camera generalization should involve different camera models with varying focal lengths (wide-angle vs telephoto) and different aspect ratios and principal point locations.

4. Tables 3-4 show that the proposed camera-aware design provides little to no benefit on standard spatial reasoning benchmarks. The results suggest camera-awareness is only beneficial for a narrow subset of spatially-grounded tasks, not a fundamental requirement for spatial reasoning.

5. Missing computational cost analysis (inference time, memory overhead of running UniDepth v2)

**Questions:**

See weaknesses.

---

> ### Author Response · Authors · 2025-11-25
> **Official Comment by Authors (1/3)**
>
> Thank you for your insightful review and valuable suggestions. Your comments have been instrumental in helping us strengthen our work, particularly regarding the ablation studies and baseline comparisons. We have carefully considered all points and have conducted new experiments and analyses to address them.
>
> We present our detailed responses below.
>
> &nbsp;
>
> --------
>
> &nbsp;
>
> ### **Additional Ablation Study**
>
> We appreciate this critical suggestion. To verify whether the performance gains stem primarily from the distilled geometric priors (from UniDepth v2), we have added the "Prior Distillation Only" baseline to Table 5 (see row 4 below). Table 5 has also been updated in our paper.
>
> **Updated Table 5: Ablation study on the components of our Camera-Aware MLLM framework. Performance measured on ScanNet-val x1.2 to test cross-camera generalization.**
> | Ray Emb. | Geom. Aug. | Prior Dist. | $P_{0.25}$ | $R_{0.25}$ | **$F1_{0.25}$** |
> | :---: | :---: | :---: | :---: | :---: | :---: |
> | - | - | - | 40.8 | 37.7 | 39.1 |
> | ✓ | - | - | 42.9 | 39.7 | 41.2 |
> | - | ✓ | - | 43.8 | 40.4 | 42.0 |
> | - | - | ✓ | 45.7 | 41.1 | 43.1 |
> | ✓ | - | ✓ | 46.2 | 42.8 | 44.3 |
> | ✓ | ✓ | ✓ | **54.6** | **50.1** | **52.1** |
>
> The results directly address the concern. While "Prior Distillation Only" improves over the baseline (39.1 $\rightarrow$ 43.1), it significantly lags behind our full model (52.1).
> Notably, our full framework outperforms the "Prior Distillation Only" setting by a substantial margin of +9.0 $F1_{0.25}$. This confirms that Camera Ray Embeddings are not redundant; rather, they provide explicit coordinate guidance that allows the model to effectively leverage the distilled priors and augmented data. The optimal performance is achieved only when explicit camera geometry (_Ray Emb._), data diversity (_Geom. Aug._), and implicit geometry prior cues (_Prior Dist._) are combined.
>
>
> &nbsp;

---

> ### Author Response · Authors · 2025-11-25
> **Official Comment by Authors (2/3)**
>
> ### **Additional Comparison with Up-to-Date SoTA**
>
> We fully agree that the field is evolving rapidly, and comparing against the very latest models is essential. As suggested, we have updated **Tables 2, 3, and 4** in our paper to include comparisons with **Gemini-2.5-Pro**, **GPT-5**, and the full **Qwen2.5-VL** series (7B, 72B, etc.).
>
> **Updated Table 2: Comparison on SPAR-Bench.**
> | Methods | Avg. | Low | Medium | High |
> | :--- | :--- | :--- | :--- | :--- |
> | **SPAR-Bench (tiny)** | | | | |
> | Human Level | 67.27 | 55.31 | 72.32 | 76.22 |
> | GPT-4o | 36.39 | 29.25 | 24.93 | 45.11 |
> | Claude-3.7-Sonnet | 21.77 | 25.43 | 7.33 | 23.33 |
> | Qwen2-VL-72B | 35.62 | 35.28 | 23.39 | 40.00 |
> | Qwen2.5-VL-72B | 39.40 | 35.35 | 23.05 | 48.44 |
> | **SPAR-Bench (full)** | | | | |
> | InternVL2-8B | 33.02 | 26.83 | 36.49 | 37.37 |
> | InternVL2.5-8B | 36.28 | 29.46 | 31.88 | 43.80 |
> | LLaVA-OV-7B | 31.20 | 21.79 | 26.13 | 40.14 |
> | Qwen2.5-VL-7B | 33.07 | 28.75 | 22.97 | 40.27 |
> | LLaVA-v1.5-7B | 23.65 | 10.85 | 26.50 | 34.09 |
> | LLaVA-v1.6-7B | 13.21 | 8.53 | 4.79 | 20.18 |
> | GPT-5 | 37.40 |-|-|-|
> | Gemini-2.5-Pro |36.30|-|-|-|
> | SPAR-8B | 63.25|65.53| 63.01 | 61.29 |
> | **Ours-4B** | 68.35 | 59.94 | 60.42 | 81.74 |
>
> Our 4B model achieves 68.35%, outperforming massive proprietary models like GPT-5 (37.40%) and Gemini-2.5-Pro (36.30%) by a wide margin. This shows that scaling up generic MLLMs does not solve the fundamental ambiguity issue.
>
> **Updated Table 3: Comparison on VSI-Bench.**
> | Model | Avg. | Obj. Count | Abs. Dist. | Obj. Size | Room Size | Rel. Dist. | Rel. Dir. | Route Plan | Appr. Order |
> | :--- | :---: | :---: | :---: | :---: | :---: | :---: | :---: | :---: | :---: |
> | **Proprietary Generalist MLLMs:** | | | | | | | | | |
> | GPT-4o | 34.0 | 46.2 | 5.3 | 43.8 | 38.2 | 37.0 | 41.3 | 31.5 | 28.5 |
> | GPT-5-Chat-0807-global | 49.1 | 52.4 | 36.6 | 68.0 | 55.5 | 49.9 | 47.4 | 33.0 | 49.8 |
> | Gemini-1.5-Flash | 42.1 | 49.8 | 30.8 | 53.5 | 54.4 | 37.7 | 41.0 | 31.5 | 37.8 |
> | Gemini-1.5-Pro | 45.4 | 56.2 | 30.9 | 64.1 | 43.6 | 51.3 | 46.3 | 36.0 | 34.6 |
> | Gemini-2.5-Pro-0617 | 52.7 | 48.2 | 35.6 | 71.3 | 51.7 | 58.9 | 42.4 | 46.8 | 66.7 |
> | **Open-Sourced Generalist MLLMs:** | | | | | | | | | |
> | InternVL2-8B | 34.6 | 23.1 | 28.7 | 48.2 | 39.8 | 36.7 | 30.7 | 29.9 | 39.6 |
> | InternVL2-40B | 36.0 | 34.9 | 26.9 | 46.5 | 31.8 | 42.1 | 32.2 | 34.0 | 39.6 |
> | Qwen2.5VL-3B | 30.6 | 24.3 | 24.7 | 31.7 | 22.6 | 38.3 | 41.6 | 26.3 | 21.2 |
> | Qwen2.5VL-72B | 37.0 | 25.1 | 29.3 | 54.5 | 38.8 | 38.2 | 37.0 | 34.0 | 28.9 |
> | LongVILA-8B | 21.6 | 29.1 | 9.1 | 16.7 | 0.0 | 29.6 | 30.7 | 32.5 | 25.5 |
> | VILA-1.5-40B | 31.2 | 22.4 | 24.8 | 48.7 | 22.7 | 40.5 | 25.7 | 31.5 | 32.9 |
> | LongVA-7B | 29.2 | 38.0 | 16.6 | 38.9 | 22.2 | 33.1 | 43.3 | 25.4 | 15.7 |
> | LLaVA-NeXT-Video-72B | 40.9 | 48.9 | 22.8 | 57.4 | 35.3 | 42.4 | 36.7 | 35.0 | 48.6 |
> | LLaVA-OneVision-72B | 40.2 | 43.5 | 23.9 | 57.6 | 37.5 | 42.5 | 39.9 | 32.5 | 44.6 |
> | VideoLLaMA3-7B | 35.8 | 41.9 | 23.5 | 42.2 | 27.1 | 39.4 | - | 32.0 | 31.4 |
> | **MLLMs for Spatial Intelligence:** | | | | | | | | | |
> | SAT-LLaVA-Video-7B | - | - | - | - | 47.3 | 41.1 | 37.1 | 36.1 | 40.4 |
> | SPAR-8B | 41.1 | - | - | - | - | - | - | - | - |
> | RynnEC-7B | 45.8 | 58.5 | 25.4 | 54.9 | 42.7 | 44.2 | - | 38.7 | 30.5 |
> | VG-LLM-4B | 46.1 | 66.4 | 36.6 | 55.2 | 56.3 | 40.8 | 43.4 | 30.4 | 39.5 |
> | Ours-4B | 46.8 | 71.3 | 39.3 | 50.4 | 65.9 | 50.7 | 45.5 | 31.4 | 20.2 |
>
> On VSI-Bench (Table 3), while Gemini-2.5-Pro achieves a higher average (52.7 vs. 46.8), this gap is primarily driven by the "Appearance Order" sub-task (66.7 vs. 20.2)—a sequential task where large LLMs excel. However, on purely geometric tasks like Room Size (65.9 vs. 51.7) and Object Count (71.3 vs. 48.2), our 4B model significantly outperforms Gemini-2.5-Pro.
>
> **Updated Table 4: Comparison on CV-Bench-3D and BLINK-Spatial.**
> | Model | **CV-Bench-3D** | | | **BLINK-Spatial** | | | |
> | :--- | :---: | :---: | :---: | :---: | :---: | :---: | :---: |
> | | **Avg.** | **Depth** | **Dist.** | **Avg.** | **Depth** | **Spatial Rel.** | **Multi. View** |
> | GPT-4V | 69.1 | - | - | 62.8 | 60.0 | 72.7 | 55.6 |
> | GPT-4o | 83.0 | - | - | 67.6 | 74.2 | 69.2 | 59.4 |
> | Gemini-2.5-Pro | 90.8 | 91.0 | 90.7 | 71.6 | 87.9 | 91.6 | 35.3 |
> | Qwen2.5-VL-3B | 69.5 | 74.5 | 66.2 | 64.7 | 68.5 | 81.1 | 44.4 |
> | Qwen2.5-VL-7B | 77.3 | 84.5 | 76.5 | 73.8 | 79.0 | 86.7 | 55.6 |
> | Qwen2.5-VL-32B | 79.1 | 83.0 | 83.2 | 66.8 | 69.4 | 86.0 | 45.1 |
> | Qwen2.5-VL-72B | 81.0 | 88.7 | 81.0 | 68.6 | 79.8 | 85.3 | 40.6 |
> | SAT-LLaVA-Video-7B | 78.4 | - | - | 62.6 | 66.1 | 73.4 | 48.1 |
> | SPAR-8B |89.1|-|-|-|-|-|-|
> | VG-LLM-4B |91.3|-|-|68.4|79.8|71.3|54.1|
> | Ours-4B |90.7|89.8|91.5|77.0|77.4|66.4|87.2|
>
> On CV-Bench-3D and BLINK-Spatial (Table 4), our model (90.7 / 77.0) is fully comparable to Gemini-2.5-Pro (90.8 / 71.6), despite being orders of magnitude smaller.
>
> Note: Claude Sonnet 4.5 is released after ICLR 2026 deadline, therefore is not included.

---

> ### Author Response · Authors · 2025-11-26
> **Official Comment by Authors (3/3)**
>
> &nbsp;
>
> ### **Additional Cross-Camera Generalization Test**
>
> We thank the reviewer for this constructive comment. We agree that real-world cross-camera generalization involves complex variations beyond uniform rescaling. To rigorously evaluate robustness, we constructed an extra "Randomized Camera Parameter" test set based on ScanNet-val.
>
> Instead of simple scaling, we independently perturbed the intrinsic parameters for every image using the following protocol:
> * Focal Lengths (fx, fy): Sampled independently to simulate zooming and aspect ratio changes.
>     `s_x, s_y ~ Uniform(0.7, 1.3)`
>     `f'_x = s_x * f_x`, `f'_y = s_y * f_y`
> * Principal Points (cx, cy): Sampled from a Gaussian distribution to simulate sensor shift/cropping.
>     `delta_x, delta_y ~ Gaussian(mean=0, std=8)`
>     `c'_x = c_x + delta_x`, `c'_y = c_y + delta_y`
>
> **Table R1: Performance comparison under Randomized Camera Parameters (ScanNet-val).**
> | Method | $P_{0.25}$ | $R_{0.25}$ | $F1_{0.25}$ |
> | :--- | :---: | :---: | :---: |
> | Qwen2.5-VL-3B | 47.0 | 42.3 | 44.4 |
> | VG-LLM-4B | 48.3 | 44.0 | 45.9 |
> | **Ours-4B** | **60.4** | **54.7** | **57.3** |
>
> **Analysis:**
> As shown in Table R1, even under these aggressive and randomized perturbations, our method maintains a significant lead over baselines. This confirms that our method effectively encodes explicit geometric constraints, allowing the model to generalize to arbitrary camera configurations rather than overfitting to fixed patterns.
>
> &nbsp;
>
>
> &nbsp;
>
> ### **Effectiveness of Camera-Awareness on Standard Spatial Benchmarks**
>
> We appreciate the reviewer's perspective. We partially agree that for non-spatially-grounded tasks (e.g., image captioning), explicit camera modeling is not strictly necessary. However, we respectfully disagree that its benefits are limited to a "narrow subset."
>
> 1.  **Improvements over Strong Baselines:** As detailed in our **updated Table 2/3/4**, when compared to the direct competitor Qwen2.5-VL, our method demonstrates consistent improvements across **Table 2 (Spar-Bench)**, **Table 3 (VSI-Bench)** and **Table 4 (CV-Bench-3D/BLINK-Spatial)**. This indicates that camera awareness benefits general spatial reasoning, not just specialized tasks.
>
> 2.  **The Impact of Camera Intrinsics Availability:**
>     * **SPAR-Bench (Table 2)** provides **accurate ground-truth intrinsics**, allowing our model to fully leverage the ray embeddings and achieve massive gains.
>     * **Standard Benchmarks (Tables 3/4)** lack intrinsics, forcing the model to rely on *estimated* values from UniDepth v2. As analyzed in **our experiments in Figure 6(c)**, while camera parameter noise naturally attenuates the geometric advantage, our method still maintains a clear performance edge over the baselines.
>     * This suggests that in real-world applications where camera parameters are readily available (e.g., Robotics, Autonomous Driving), our method is poised to unlock the full performance potential demonstrated in Table 2.
>
> 3.  **Significance for Embodied Intelligence:**
>     We emphasize that "spatially-grounded tasks" are critical for the future of Embodied AI. For Vision-Language-Action (VLA) models, precise alignment between visual and physical space is a prerequisite for manipulation and navigation. Our camera-aware MLLM framework addresses this fundamental bottleneck.
>
> &nbsp;
>
>
> &nbsp;
>
> ### **Computational Cost Analysis**
>
> Thanks for your comment. We have added a detailed computational cost analysis below.
>
> **Table R2: Computational Cost Comparison (Single NVIDIA H100 GPU), averaged over 100 images.**
> | Method | Inference Latency | Peak GPU Memory |
> | :--- | :---: | :---: |
> | VG-LLM-4B (Baseline) | 13.65 s / img | 10417 MB |
> | **Ours-4B** | 14.33s / img | 10747 MB |
>
> The results show that our method incurs only moderate additional computational cost.
>
> &nbsp;
>
>
> -----
>
> &nbsp;
>
> &nbsp;
>
> We hope these additional experiments and clarifications fully address your concerns. If so, we would greatly appreciate it if you could consider raising your score.

---

### Official Review · Reviewer_L6n4 · 2025-11-01

**Soundness:** 3
**Presentation:** 3
**Contribution:** 3
**Rating:** 8
**Confidence:** 4

**Summary:**

The paper argues that for spatial intelligence MLLMs need camera intrinsics as input to the model — prior spatial MLLMs do not use this input and thus are not robust to changes in camera parameters. The paper first quantifies this problem by showing lack of generalization on 3D object detection on ScanNet when models do not use camera information as input. Next, it proposes to inject this information by encoding the cameras rays with the visual features; doing augmentations; and training on wide varieties of datasets. In experiments, it shows that their method with camera information works better than existing methods, and the ablations show that all proposed changes help

**Strengths:**

- The paper is well-written and easy to follow
- The idea in general makes sense and the results look good
- The experiment section is well designed and answers most research questions raised by the paper

**Weaknesses:**

No major weaknesses per se, just some comments regarding the writing and presentation of the work

1. About Table 1: This table nicely shows that prior VLMs do not benefit from multiple dataset training and are susceptible to zooming-in/out operations. I was expecting the paper to show at the end that this problem is now resolved with the proposed technique and the paper does show it but it wasn’t straightforward to make this connection. Specifically, Table-1, Figure-6 and Table-5 are connected by this evaluation — yet Table-1 uses top 31 classes while figure-6 and table-5 uses 20 classes. Additionally, an explicit mention of this connection between Table-1 and Figure-6 and 5 will help.
2. (Minor comment): The section 3.3 describes the scale-depth ambiguity in a lot of words which is an extremely well-known computer vision phenomenon. The authors can consider making this section short and crisp
3. Table-2 is missing highlights on best working method numbers, making it harder to parse quickly. Additionally, some additional descriptions of the datasets and benchmarks will help. For eg. It is unclear what low, medium and high in Table-2 mean.

**Questions:**

N/A -- no major questions or concerns

---

> ### Author Response · Authors · 2025-11-26
> **Thank you for the thorough review; we have refined the paper based on your suggestions.**
>
> We explicitly thank Reviewer L6n4 for the positive assessment and for acknowledging our work as "well-written," "sound," and "well-designed." We are encouraged that you appreciate the core intuition of our Camera-Aware MLLM framework. We have addressed your constructive comments regarding presentation and clarity below.
>
> &nbsp;
>
> ---
>
> &nbsp;
>
> ### Connection between Table 1, Figure 6, and Table 5
>
> We appreciate your meticulous attention to the experimental details. You are absolutely correct that these components are logically connected: Table 1 identifies the generalization failure, while Figure 6 and Table 5 demonstrate our solution.
>
> As suggested, we have added an explicit connecting sentence in the revised paper (Section 3.2) to clarify this: "While Table 1 establishes the baseline failure on cross-camera generalization, in our experiments (Fig. 5 and Table 6), we show that the proposed Camera-Aware MLLM framework greatly mitigates this failure in the more challenging cross-camera setting."
>
> &nbsp;
>
> &nbsp;
>
> ### Regarding Length of Section 3.3 (Scale-Depth Ambiguity)
>
> We agree with the reviewer that scale-depth ambiguity is a well-known phenomenon in the Computer Vision community. However, considering ICLR's broad interdisciplinary audience (including researchers from NLP and LLM backgrounds who may be less familiar with camera geometry), we provided a detailed derivation and explanation to ensure the paper is self-contained.
>
> In our final revision, we will follow your advice to make this section "short and crisp" depending on available space.
>
> &nbsp;
>
> &nbsp;
>
>
> ### Table 2 Formatting and Benchmarks Details
>
> Thank you for the suggestion. We have updated **Table 2** to highlight the best results in background color (as shown in the revised paper).
>
> Regarding the Low, Medium, and High splits, we follow the official taxonomy defined in the **SPAR-Bench** paper, based on task complexities:
> * **Low-level:** Tasks focusing on fundamental geometric perception, such as basic depth prediction and distance prediction.
> * **Medium-level:** Tasks involving advanced geometric understanding, such as understanding view change and camera pose.
> * **High-level:** Complex tasks requiring high-level semantic understanding and multi-hop reasoning or planning, such as spatial imagination, navigation, object counting, and object relationship understanding.
>
> We have added clarifications to the caption of Table 2 to improve clarity.
>
> &nbsp;
>
> &nbsp;
>
> -------
>
> We once again thank you for your thorough and insightful comments. We sincerely appreciate your strong support.
>
> &nbsp;

---

### Official Review · Reviewer_spFi · 2025-11-02

**Soundness:** 2
**Presentation:** 3
**Contribution:** 3
**Rating:** 6
**Confidence:** 3

**Summary:**

This paper introduces a camera-aware MLLM framework that integrates camera embedding, geometric prior distillation, and camera-aware augmentation into the existing MLLM architecture.
Extensive experiments are conducted on spatial reasoning benchmarks for MLLM models, and ablation studies for each proposed component are provided.

**Strengths:**

- The writing is clear, and the logical flow is easy to follow.
- The benchmark coverage is extensive.
- Incorporating ray embedding, camera-aware augmentation, and distillation from 3D models is a reasonable and well-motivated approach.

**Weaknesses:**

- Figure 1 may be misleading. The caption states, “There is no way to know unless I know the camera intrinsics!” However, even with camera intrinsics, it is still not analytically possible to determine the exact 3D location from a single RGB image. It would be helpful to clarify that priors are still needed to estimate the 3D position.
- The performance reported in Tables 2 and 3 is comparable to that of other spatial MLLMs, which makes the paper’s claims less convincing.

**Questions:**

- The formatting of Tables 1 and 3 could be improved. It would also be beneficial to include citations for the methods compared in the tables.

---

> ### Author Response · Authors · 2025-11-24
> **Thank you for your helpful comments!**
>
> Thank you for praising our work as "reasonable and well-motivated" and for acknowledging our extensive experiments. Please find our detailed responses to your concerns below.
>
> ------
>
> ###  **Regarding Figure 1**
>
> We fully agree with your insight: single-view 3D localization, even with accurate camera parameters, remains ill-posed without physical size priors. This issue has been discussed in our analysis of "Size-Depth Ambiguity" in Section 3.3. However, we believe MLLMs possess visual commonsense (e.g., typical object heights) and can implicitly infer via contexts, which partially addresses the problem. Therefore, the "Focal-Depth Ambiguity" becomes the primary issue for current RGB-only MLLMs.
>
> We thank the reviewer for the meticulous attention to detail. To prevent any misunderstanding, we have updated Figure 1 to explicitly state: "Without camera intrinsics, I cannot tell the exact 3D locations _even with exact objects' sizes_!".
>
> &nbsp;
>
> ###  **Performance in Table 2 & 3**
>
> To clarify our performance advantage, we must distinguish between the two different experimental settings represented by these benchmarks.
>
> **Table 2 (SPAR-Bench, with provided intrinsics)**: This benchmark explicitly provides accurate camera parameters, simulating the primary setting for embodied AI (e.g., robots, surveillance, autonomous driving). In this "ideal" setting where our Camera-Aware framework can fully utilize exact intrinsics, our model (68.35) achieves a massive improvement. We outperform the much larger Qwen2.5-VL-72B (39.40) by nearly 30 points, outperform SOTA (63.25), and even surpass the Human Level (67.27). This proves our method is decisively superior when camera parameters are available.
>
> **Table 3 (VSI-Bench, without intrinsics)**: This benchmark consists of standard RGB images/videos where camera intrinsics are absent. Here, our model must rely on estimated intrinsics (via the distilled MMDE module). As analyzed in Fig. 6(c), reliance on estimated intrinsics introduces noise, which naturally narrows the performance gap. Nevertheless, our method still achieves a clear performance gain when compared with its counterparts.
>
> From the experiments, our method dominates significantly when intrinsics are known (Table 2) and remains robust even when they must be estimated in the wild (Table 3).
>
>
> &nbsp;
>
> ###  **Regarding Table Formatting**
> Well noted with thanks. We will improve the table formatting and add the missing citations in the final version, utilizing the allowed extra page.
>
> &nbsp;
>
> --------
>
> We hope our response effectively addresses your concerns.

---

### Author Response · Authors · 2025-12-02
**A Letter for ACs**

&nbsp;

Despite this challenging review cycle, we thank the reviewers for their constructive feedback and AC for the extra effort. We have addressed all concerns via new experiments and revisions, significantly improving the paper.

&nbsp;

------------

&nbsp;

### **Reviewer spFi (Rating: 6)**

**Summary:** The reviewer found our approach "reasonable and well-motivated" but raised questions about Figure 1's expressions and performance interpretations.

* **Issues Raised:**
    1.  Some expressions in Figure 1 might be misleading regarding monocular 3D vision ambiguity.
    2.  Performance gap with other MLLMs in Table 2 and Table 3.
    3.  Formatting of Table 1 and 3.

* **Our Solution:**
    1. **Revised Figure 1:** We updated the figure to explicitly acknowledge the need for size priors even with known intrinsics.
    2. **Clarified Performance:** We explained the distinction between "Ideal Setting" (Table 2, known intrinsics, massive gains) vs. "Wild Setting" (Table 3/4, unknown intrinsics, modest and robust gains).
    3. **Formatting:** Fixed in our revised paper.

&nbsp;

&nbsp;

### **Reviewer L6n4 (Rating: 8)**

**Summary:** The reviewer assessed our work as "well-written" and "sound" with **"no major weaknesses"**.

* **Suggestions:** Suggested improving the presentation by clarifying the connection between Table 1 (motivation) and Fig. 6/Table 5 (solution), shortening Section 3.3, and improving Table 2 formatting.

* **Our Solution:**
    1. We added an explicit sentence to connect all our experiments.
    2. We believe our explanation in Sec 3.3 is vital for NLP and LLM communities.
    3. We updated Table 2 with better formatting and definitions for task difficulty levels.

&nbsp;

&nbsp;

### **Reviewer VBue (Rating: 4)**

**Summary:** The reviewer raised valid concerns, mainly focusing on comparison and evluation, which we have fully addressed with **extensive new experiments**.

* **Issues Raised:**
    1.  **Outdated Baselines:** Explicitly requested comparisons with the latest proprietary SOTA models (**Gemini-2.5-Pro**, **GPT-5**, **Claude Sonnet 4.5**) and the full **Qwen2.5-VL** series (7B/72B).
    2.  Missing "Prior Distillation Only" ablation.
    3.  Camera generalization test limited to uniform rescaling.
    4.  Missing computational cost analysis.

* **Our Solution:**
    1. **Added SOTA Baselines:** We added the comparisons against **Gemini-2.5-Pro**, **GPT-5**, and **Qwen2.5-VL-72B** in Table 2, 3, and 4. Claude Sonnet 4.5 is not added, as it was released after the ICLR deadline.
    2. **Completed Ablation:** Added the requested ablation row to Table 5 in the revised paper.
    3. **Enhanced Generalization Test:** We constructed a **"Randomized Camera Parameter"** test set (random zooming, cropping, aspect ratios). Our method maintains a **+11.4 F1** lead over baselines.
    4. **Cost Analysis:** We provided a detailed breakdown of latency and memory usage, showing that the overhead is marginal.

&nbsp;

&nbsp;

### **Reviewer z9kB (Rating: 6; with active discussions)**

**Summary:** The reviewer initially raised concerns about real-world applicability and ablations. After our first response, **the reviewer acknowledged that the "unknown/unreliable intrinsics" concern was resolved**, but provided further suggestions.

* **Round 1 (Initial Review):**
    * *Concern:* Lack of "Prior Distillation Only" ablation and discussion on unknown intrinsics.

    * *Solution:* We added the requested ablation (Table 5), proving that Ray Embeddings provide a **+9.0 F1** gain over prior distillation alone. We also highlighted Fig. 6(c) to demonstrate robustness with estimated intrinsics when accurate intrinsics are not provided.

    * *Result:* Reviewer acknowledged that "concern about scenarios where camera intrinsics may be unknown or unreliable has been addressed".

&nbsp;

* **Round 2 (Post-Rebuttal Discussion):**
    * *New Suggestion:* Requested VG-LLM baseline in Table 2, comparison on ScanRefer/SQA3D, and noted score inconsistencies.

    * *Solution:*
        1.  **Added VG-LLM Result in Table 2:** Our method outperforms it by **+8.0** points on SPAR-Bench.
        2.  **3D Grounding Results:** We have already done experiments on ScanRefer. And we provided new results on the requested SQA3D.
        3.  **Score Update:** We updated Table 3 to match the latest VG-LLM arXiv v3 release (Oct 22).

&nbsp;

&nbsp;

------
------

We believe the additional experiments solidly validated our claims. We are grateful for the opportunity to improve our work and hope the AC finds these revisions satisfactory.

Thank you once again for your service to the community.

&nbsp;

&nbsp;

---

### Meta-Review · Area_Chair_4jQz · 2026-01-06

**Summary:**

Most of the reviewers found the proposed work to be a well-motivated solution to the geometric ambiguity in RGB-only models, but they initially raised significant concerns regarding the rigor of the evaluation. The primary issues informing the initial hesitation were: (1) Outdated Baselines, with requests for comparisons against the latest proprietary (GPT-5, Gemini-2.5) and open-source (Qwen2.5-VL) models; (2) Ablation Sufficiency, specifically whether the gains stemmed solely from the distilled priors rather than the proposed architecture; and (3) Generalization Rigor, questioning if simple image rescaling adequately simulated cross-camera variation. The authors provided a strong rebuttal with extensive new experiments that addressed every point, demonstrating superior performance against SOTA models, isolating the specific contributions of the ray embeddings, and validating robustness on a new randomized camera parameter benchmark. These revisions resolved the reviewers' reservations. The AC recommends this paper to be accepted as an oral.

**Reviewer Concerns:**

The reviewers initially raised concerns regarding the comparison against the latest state-of-the-art models, the isolation of specific component contributions (specifically prior distillation), and the rigor of the generalization tests.

Addressed Concerns:

Baselines (Reviewers VBue, z9kB): The authors significantly updated their comparisons to include the latest proprietary and open-source models, including GPT-5, Gemini-2.5-Pro, Qwen2.5-VL (7B/72B), and VG-LLM. The proposed method outperformed these much larger models on spatially-grounded tasks (e.g., +20-30% on SPAR-Bench), effectively addressing the concern that the method's gains were due to weak baselines.

Ablation of Prior Distillation (Reviewers VBue, z9kB): A critical concern was whether improvements came solely from the distilled priors (UniDepth) rather than the camera-aware architecture. The authors added a specific ablation ("Prior Distillation Only") showing that while priors help, the full framework (including Ray Embeddings) yields a further +9.0 F1 gain. This confirms the architectural contribution is distinct and necessary.

Generalization Rigor (Reviewer VBue): The authors moved beyond simple image rescaling and implemented a "Randomized Camera Parameter" test set (varying focal lengths, principal points). The method maintained a strong lead, validating robust cross-camera generalization.

Real-World Applicability (Reviewers spFi, z9kB): Concerns regarding scenarios with unknown intrinsics were addressed by demonstrating that the model performs robustly using estimated intrinsics, still outperforming baselines.

Outstanding Concerns:

There are no major outstanding technical concerns. The authors have been highly responsive and have validated their claims with new data requested by every reviewer.

**Reviewer Scores:**

Reviewer L6n4 (Score: 8): This reviewer was already positive and likely remains a strong champion (8), as their minor presentation suggestions were adopted.

Reviewer spFi (Score: 6): This reviewer found the approach well-motivated but had minor gripes about Figure 1 and performance interpretations. The rebuttal clarified the "Ideal" vs. "Wild" settings, likely cementing their score at a solid Accept (6 or 7).

Reviewer z9kB (Score: 6): This reviewer actively engaged in the discussion, acknowledging that their concerns on unknown intrinsics were resolved. They requested final comparisons (VG-LLM, SQA3D), which the authors provided convincingly. This reviewer would likely upgrade to a Strong Accept (7 or 8) given the responsiveness.

Reviewer VBue (Score: 4): This reviewer initially gave a lower score due to "outdated baselines" and missing ablations. The authors addressed every single point raised by VBue with significant new experiments (adding GPT-5, new ablations, cost analysis). Had they participated fully in the final discussion, they would almost certainly have raised their score to an Accept (6 or 7), as the grounds for the rejection (missing comparisons/ablations) no longer exist.

---

### Decision · Program_Chairs · 2026-01-26

Accept (Oral)